# Hough Transform Proposal and Simulations for Particle Track Recognition for LHC Phase-II Upgrade

**DOI:** 10.3390/s22051768

**Published:** 2022-02-24

**Authors:** Alessandro Gabrielli, Fabrizio Alfonsi, Francesca Del Corso

**Affiliations:** 1INFN Bologna, Viale Berti Pichat 6/2, 40127 Bologna, Italy; fabrizio.alfonsi@bo.infn.it (F.A.); francesca.delcorso@bo.infn.it (F.D.C.); 2Physics and Astronomy Department, University of Bologna, 40126 Bologna, Italy

**Keywords:** Hough transform, high energy physics, particle recognition, FPGAs

## Abstract

In the near future, LHC experiments will continue future upgrades by overcoming the technological obsolescence of the detectors and the readout capabilities. Therefore, after the conclusion of a data collection period, CERN will have to face a long shutdown to improve overall performance, by updating the experiments, and implementing more advanced technologies and infrastructures. In particular, the largest LHC experiment, i.e., ATLAS, will upgrade parts of the detector, the trigger, and the data acquisition system. In addition, the ATLAS experiment will complete the implementation of new strategies, algorithms for data handling, and transmission to the final storage apparatus. This paper presents an overview of an upgrade planned for the second half of this decade for the ATLAS experiment. In particular, we show a study of a novel pattern recognition algorithm used in the trigger system, which is a device designed to provide the information needed to select physical events from unnecessary background data. The idea is to use a well known mathematical transform, the Hough transform, as the algorithm for the detection of particle trajectories. The effectiveness of the algorithm has already been validated in the past, regardless of particle physics applications, to recognize generic shapes within images. On the contrary, here, we first propose a software emulation tool, and a subsequent hardware implementation of the Hough transform, for particle physics applications. Until now, the Hough transform has never been implemented on electronics in particle physics experiments, and since a hardware implementation would provide benefits in terms of overall Latency, we complete the studies by comparing the simulated data with a physical system implemented on a Xilinx hardware accelerator (FELIX-II card). In more detail, we have implemented a low-abstraction RTL design of the Hough transform on Xilinx UltraScale+ FPGAs as target devices for filtering applications.

## 1. Introduction

The High-Energy Physics Experiments (HEPE) at the Large Hadron Collider (LHC) at CERN in Geneva are approaching a high-luminosity Phase-II upgrade [1]. In an acquisition process, the data are received from the detectors and compose the physical Events [2]. In case these data belong to a certain particle interaction, within a certain time interval, they must be fully collected and set aside from the background. The Tracker [3,4], which is part of a more complex detection, and pattern recognition system, is the apparatus that reads the detectors closest to the interaction point, and reconstructs the particle tracks. In this environment, the Trigger [5] electronic system selects the useful data from background and, in general, this task of selecting a few expected events, from a large amount of unnecessary Events, is performed through a pattern recognition process. Here, this recognition task is studied using the Hough transform (HT) [6,7] and aims at the latest generation of FPGAs for final hardware implementation. Furthermore, as it is impossible to test the hardware before the system is complete, some components need to be validated before assembly. In fact, we have demonstrated through post-layout simulations and performance comparisons that the HT approach can fit on a single FPGA. The main advantage of using FPGAs, over other programmable components, is their speed and low latency to handle pattern recognition problems, also in a noisy environment. FPGAs provide latency that increases linearly with incoming data, while other solutions increase latency time more quickly. Furthermore, here, the noise can be imagined of any origin, and therefore composed of traditional white noise or unwanted physical data that emerge after particle collisions. In both cases, this unwanted data must be removed from the useful information, and HT appears to be a reasonable solution for the purpose.

## 2. Pattern Recognition in HEPE Using the Hough Transform

Recently, the HT algorithm has been studied and implemented on many electronic cards [8,9,10,11]. In particular, in this study we show the algorithm investigated for tracking detectors in HEPE, where it is used to extract trajectories and straight or curved lines, within digital image representations of the particle space. This space, in this study, is the transversal plane with respect to the beam line, where the pairs (r, φ) identify a potential physical pixel ignited by a charged particle. These points are called Hits. Hence, the HT, which is a mathematical transformation, turns this particle space into a space of parameters called Hough space (HS).
(1)$$\frac{Aq}{Pt}=\frac{\phi_{0}-\varphi }{r}$$

Formula (Equation 1) executes the conversion from the particle space to the HS using (r) as the radius of the detector layer in the real transversal plane, (φ) as the angle of the Hit with respect to the horizontal axis; let us say that these are the input parameters. On the contrary, ($\phi_{0}$) is the same angle of the particle trajectory with respect the origin, and let us call it an output parameter of Formula (Equation 1), along with the ($\frac{Aq}{Pt}$), which is described below (see [8,9] for details). Formula (Equation 1) computes, for each pair of Hits, many operations by varying ($\phi_{0}$) angles and, consequently, any Hit in the parameter space creates many straight-lines in the HS. This process, carried out over the entire set of input Hits, i.e., the Event, is here defined as Forward Computation.

### 2.1. Forward Computation

Figure 1 shows two straight-lines in the HS, created by processing 5 different ($\phi_{0}$) angles over 2 constant pairs of (r, φ) parameters (Hits). Hence, 2 groups of 5 *green* and *red* points are generated by executing 10 times the Formula (Equation 1). This is why, as previously mentioned, that a point in the parameter space generates a straight-line in the HS. Furthermore, the two straight-lines in the HS cross in one common point, the orange point, and this particular point shares the ($\phi_{1}$) angle. This cross point is below called Road.

The HS is a space parametrized by the ($\frac{Aq}{Pt}$, $\phi_{0}$) pairs. The Hits, parametrized by (r, ϕ), are in correspondence with the ($\frac{Aq}{Pt}$, $\phi_{0}$) points in the HS.

In Formula (Equation 1), ($\frac{Aq}{Pt}$) represents the particle momentum (Pt), measured in kg·m/s. The charge (q) is measured in Coulomb, and the parameter (A) is a constant factor to turn the ($\frac{Aq}{Pt}$) units in rad/m. The points in the HS compose the straight straight-lines generated by the *Forward Computation* process via the execution of the Formula (Equation 1), as many times as the ($\phi_{0}$) bins are defined. Figure 2 shows a simulation of an entire Event transposed to the HS. The cross points indicate the identification of potential Roads, and in case these Roads are confirmed, the Hits that have generated these Roads are marked as belonging to the same track.

### 2.2. Physics Case

We created a Python emulator to validate the behaviour and the performance of a physical system. In this way, the software and firmware designs were compared using the same input data. The system performance is measured by the capability of recognizing the input tracks, i.e., the number of tracks detected compared to the total number of expected tracks. In this study we have emulated a sector of the ATLAS tracker composed of Pixel and Strip detectors, and we have estimated the pattern recognition capability using the following specifications:Eight input channels covering part of the Pixel [12] and part of the Strip Detector [13] of the ATLAS experiment;Up to 1000 input Hits, representing tracks and background altogether, used for the (*r*, ϕ) pairs, digitized respectively with 12 and 16 bits. The background is evenly distributed over the channels;Input Hits computed as many times as ($\phi_{0}$) bins there are in the HS, and hence up to 1200 bins, which are digitized with 11 bits;A clock frequency targeted at 250 MHz for the final FPGA implementation;($\frac{Aq}{Pt}$) inverse momenta, used in Formula (Equation 1), binned with up to 216 bins, digitized with 8 bits.

Figure 3 shows a reduced size example of a 3D histogram representation of all the straight-lines generated via the HT Formula (Equation 1). The Accumulator circuit is a component designed to handle the HS space, and it is composed of an electronic memory dimensioned for 1200 columns, which are the ($\phi_{0}$) angles, and 64 rows, which are the ($\frac{Aq}{Pt}$) inverse momenta.

If a ($\phi_{0}$,$\frac{Aq}{Pt}$) pair indicates a potential physical Event, it is saved as candidate Road, and marked as ($\phi_{0}$,$\frac{Aq}{Pt}$)${}_{Road}$. This latter pair is one of the cross points in the HS, and it is shown in Figure 1 as an orange point, in Figure 2 as cross points of many straight-lines, and in Figure 3 as the unique 8 high column of the 3D histogram. This 3D image is a graphical representation of the Accumulator memory. It shows the number of input channels each bin has been crossed by the straight-lines generated via the *Forward Computation*. In the plot, the bin numbers range from 0 to 8, for the 8 input channels, and the unique cell 8 high identifies the Road.

For example, if a Hit that belongs to the 7th layer gives rise to a straight-line which crosses the n-th row, and the m-th column of the Accumulator, then the bin addressed at n-th row—of the 64 ($\frac{Aq}{Pt}$)—and at the m-th column—of the 1200 ($\phi_{0}$)—is updated in its 7th floor. This process is carried out in parallel, within every clock cycle, for up to 1200 ($\phi_{0}$) angles, and for 8 inputs, which is why it requires a large amount of electronic resources to fit into an FPGA.

### 2.3. Backward Computation

Once the Accumulator has been filled up with all the straight-lines generated by the *Forward Computation*, the selection of candidate Roads starts, and some ($\phi_{0}$,$\frac{Aq}{Pt}$${)}_{Road}$ pairs can be found. Then, for each candidate Road all the input Hits must be re-considered to search which ones originated from that particular Road. The HT Formula (Equation 1) is executed backward, in a reverse mode, and this process is called *Backward Computation*.
(2)$${\left(\frac{Aq}{Pt}\right)}_{Road}-\frac{Aq}{Pt}={\left(\frac{Aq}{Pt}\right)}_{Road}-\left(\frac{{\phi_{0}}_{Road}-\varphi }{r}\right)$$

Formula (Equation 2) shows that the (ϕ${}_{0}$${}_{Road}$) value and the Hit (*r*, ϕ) numbers are considered to calculate the ($\frac{Aq}{Pt}$) value first, and then compare it with the (${\frac{Aq}{Pt}}_{Road}$). The comparison, in principle, should result in a perfect match between ($\frac{Aq}{Pt}{)}_{Road}$) and ($\frac{Aq}{Pt}$). In case of a perfect match, the (r, ϕ) numbers are saved and sent to output as Hits belonging to that Road. On the contrary, in case the Formula (Equation 2) does not match, the (*r*, ϕ) numbers are discarded. This operation is done in parallel for the entire set of Hits, which again is why it requires a large amount of electronic resources to fit into an FPGA. Figure 4 shows a sketch of the *Forward* and *Backward Computations*.

## 3. HT Synthesis on FPGA

Firmware (FW) design can be approached at different levels of abstraction. For this case study, we used a low level of abstraction because we did not want to spread the netlist components over the whole layout area, even if the synthesis tools are now optimized for the purpose. The recent common use of high-level [14] synthesis would not give much benefit, because we need to implement, in one FPGA, the largest system in terms of the parameters listed in Section 2.2. The design optimization includes the electronic implementation of the Formula (Equation 2), the use of Digital Signal Processors (DSP), Configurable Logic Blocks (CLB), and Look-Up Tables (LUT). Leaving the automatic tools to optimize the system, without constraining where to distribute the netlist on the FPGA’s Super Logic Regions (SLR) [15] would not optimize the resource allocation. The interconnections between different SLRs are in fact slower than those within one SLR. So, we have not implemented critical high-speed nets sharing different SLRs. We adopted a FW description using the VHDL Register Transfer Logic (RTL) level of abstraction, and the code was easily simulatable. Frontier FPGA selection and low-level abstraction description is justified by the need to cope with complex circuits to fit small devices. Furthermore, the choice of FPGAs guarantees a fixed Latency, as estimated below, since this is a requirement to be fulfilled in HEPE. In addition, to match the Timing Constraints of the synthesis process, the Hold and Setup times have to be properly considered. In particular, in case of a Setup time mismatch, in general, the problem can be fixed by reducing the clock frequency, if permitted. By contrast, when Hold-Time violations appear, the circuit does not work at all, independently of the clock frequency. This is one of the main reasons why we have considered the design of critical nets first, starting from the VHDL code. In fact, parameters such as Hold-Time and Max-FanOut were set as specifications before performing any synthesis. Pipeline [16] stages have also been inserted in sequential logic to facilitate the Place and Route automatic process. Eventually, the *Backward Computation* was implemented exploiting DSP for divisions and LUTs for additions.

Moreover, this case study of the HT implementation has been focused on the FELIX-II card [17], featuring the Xilinx UltraScale+ VU9P FPGA. This FPGA uses 24 transceivers [18], which are used to read 8 channels, and to provide 16 parallel outputs. We have run simulations including parasitic capacitances extracted from the design. In this way we were also able to check out the Worst Negative Slack (WNS) for the Setup time. As a consequence of that, we also estimated the Working Frequency, the Input and Output Throughput, and the Total Latency.

Working Frequency = $\frac{1}{T_{clock}-WNS}$ GHz = $\frac{1}{4+0.5\;\mathrm{ns}}$ GHz = 0.22 GHz;Input Throughput ($\frac{\mathrm{Byte}}{s}$) = $\frac{8\;\times \;28\;\times \;250\;\mathrm{MHz}}{8\;\mathrm{bit}}$ = 7 GBps;Output Throughput ($\frac{\mathrm{Byte}}{s}$) = $\frac{16\;\times \;28\;\times \;250\;\mathrm{MHz}}{8\;\mathrm{bit}}$ = 14 GBps;Total Latency = 30 clock periods.

Furthermore, the processing time to fully analyze an Event is a function of the following parameters, besides the clock frequency:#Hits, which is the total number of input data (Hits) loaded 8 at a time;#Roads, which is the total number of identified Roads within a given Event;Latency, which is the number of clock periods from when the entire Event is loaded by executing the *Forward Computation*, to when the first group of 16 Hits, belonging to the first Road, is sent out.

## 4. Simulations

The VHDL code was simulated to first verify the behavior of the system, then a synthesis process and a consequent evaluation of the parasitic capacities allowed us to confirm the behavior, through further simulations. The same approach has been applied to build post-layout simulations, which are more accurate and closer to reality. The entire set of simulations have also been permitted to estimate and to validate the clock frequency and throughputs. As previously mentioned, in parallel to FW simulations, we have built a software emulator to run the HT process using a Python code. The software only did not consider the internal latencies and other timing templates of the electronics. In this way, we have been able to compare the two types of simulations, the FW-based, and the SW-based, as shown in Figure 5. Eventually, we have compared the outputs provided by the two simulations sharing the same inputs. The input pattern generated for the simulations—test vectors—have also been used to run a physical test on an FELIX-II Ultrascale+ [17] platform, featuring the Xilinx VU9P FPGA, as shown in Figure 6. The FW has been tested, the data have been fed, and read out through a 16-lane 3rd generation PCI bus. These tests have succeeded in confirming that the simulations were correctly performed.

Table 1 shows the simulation comparisons as they come out from individual processes that share the same inputs. The generated input Hits are dummy, contain a given number of Roads, have been analyzed in the Python simulator first and, subsequently, in the VHDL-based FW simulator. All the Events contain a common bunch of 128 noisy Hits and refer to 8 input layers shared among Pixels and Strips, differentiated by the (*r*, φ) values. In this case, each Road is composed of 2 × 8 Hits and the system uses a 216 × 216 Accumulator. In particular, Table 1 shows the number of Hits, the number of Generated Roads, and how many candidate Roads the system extracts first, and processes successively. The last column shows the entire Event processing time, from when it has been fully loaded into the FPGA to when the first Road, along with the corresponding Hit information, is sent to the output. It is evident that the HT algorithm finds more Roads with respect to those created, but this is normal, as the extracted Roads are just potential physical tracks that must be confirmed by a successive process. Since the number of real tracks entering the system is unknown, the most important thing is to not lose any of them among the background noise. If more Roads are generated, these will be eliminated by a further cross-check using additional layers than those used in the HT process. In [8,9], we report a hardware implementation, with the detail of the block diagram, while in this paper we broaden the field to generic low latency applications, and this is why we have designed a software development tool. Compared to other similar devices [19,20,21], FPGA components can simultaneously provide low and fixed latency, low power budget, and high data rate. Furthermore, the HT implementation on FPGAs is independent of input data and noise percentage. The system also becomes more efficient for large input data sets as the overall latency scales linearly.

## 5. Conclusions

FPGAs have been proved to be good devices to host mathematical algorithms as the Hough transform. In particular, the latest Xilinx families, due to the UltraScale+, featuring high numbers of embedded DSPs, memories, transceivers, and CLBs, is one of the best candidate target devices. For test purposes, we have used a custom card designed originally for the ATLAS data acquisition and trigger applications at CERN. This card, namely the FELIX-II [17], is populated with an Ultrascale+ device. The study has investigated a Hough transform system characterized by a 250 MHz clock signal, a 216 × 216 Accumulator, and a number of Hits on the order of 1000. The choice of FPGAs in this field is justified by the need of high input-output data transmission speed, low system latencies, and reasonable management of the algorithm complexity. In fact, FPGA components, with respect to other similar devices, can simultaneously provide low and fixed Latency, low power budget, and high data rate, by reducing the resource allocations of two orders of magnitude. The design process has been fully simulated from the firmware code to the final implementation. Currently, Hough’s transform firmware is finalized with an RTL description, which is synthesizable and routable on FPGA resources. Post-synthesis and post layout simulations were also performed to consider internal FPGA physical connections. In addition, a Python tool was designed with the aim of deeply investigating the behavior of the system before laboratory testing. With this tool, many dummy inputs have been prepared via a series of Hits over a noisy background to estimate the performance of the entire architecture in finding the candidate tracks. Eventually, we loaded the algorithm into a board featuring Ultrascale+ FPGA and provided the same simulated test vectors used in the simulations: the two setups matched perfectly. These studies of Hough transform algorithms for application to high-energy physics experiments are not new but, particularly for hardware particle recognition tasks in tracking systems, have never been completed using an FPGA implementation. 

## Figures and Tables

**Figure 1 sensors-22-01768-f001:**
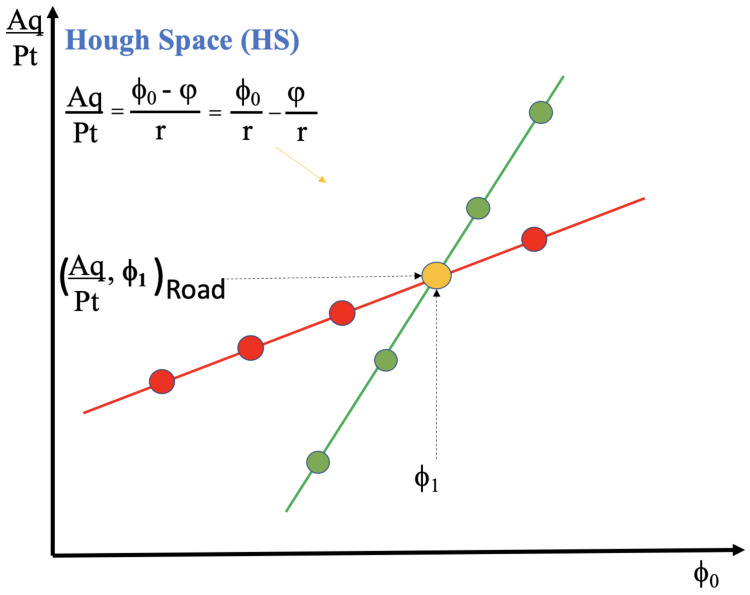
Hough space created via Formula (Equation 1).

**Figure 2 sensors-22-01768-f002:**
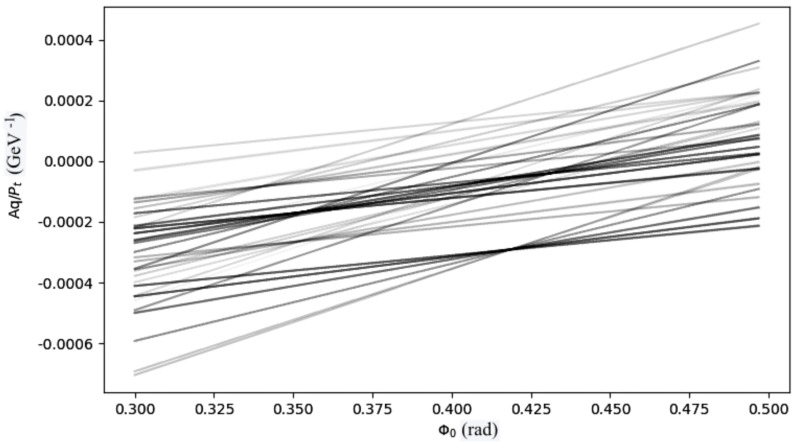
Hough space example (electron-Volt eV and charge (q) set to 1 units).

**Figure 3 sensors-22-01768-f003:**
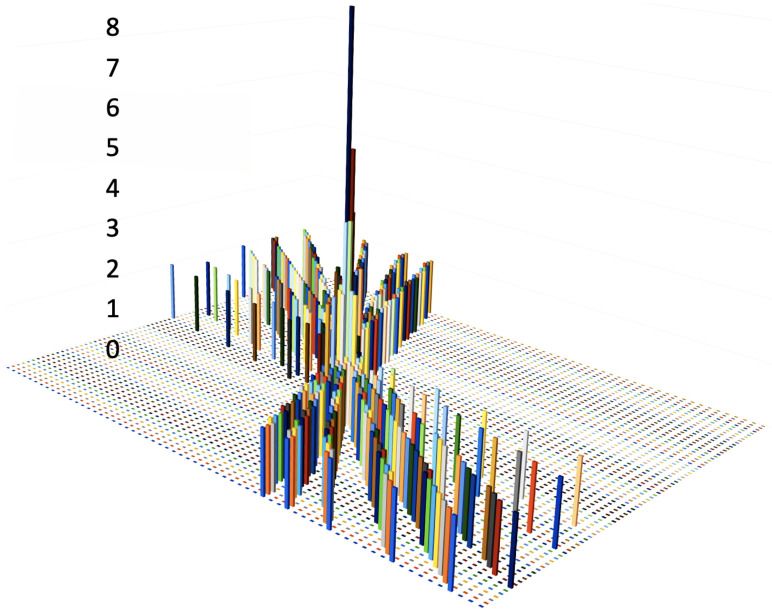
3D plot representation of the HS with a unique 8 high bin: the identified Road.

**Figure 4 sensors-22-01768-f004:**
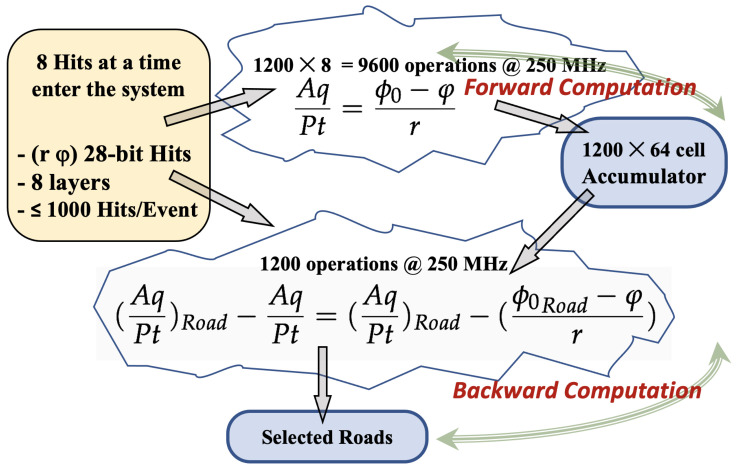
Forward and Backward Computations.

**Figure 5 sensors-22-01768-f005:**
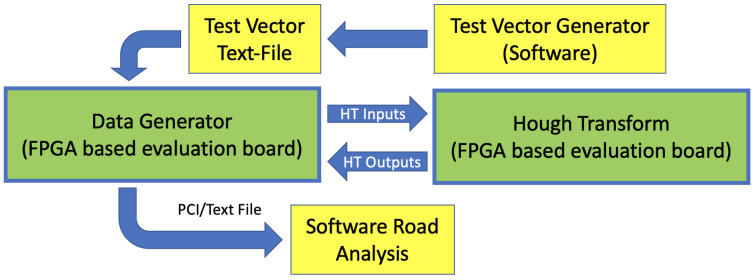
Logic description of the Hough transform demonstrator.

**Figure 6 sensors-22-01768-f006:**
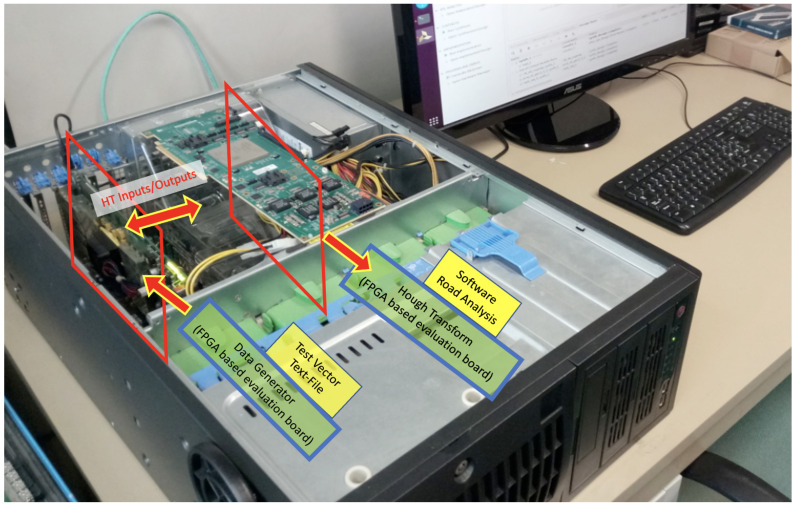
Hough transform demonstrator under test.

**Table 1 sensors-22-01768-t001:** Table summarizing the simulation results for different parameters (# means number of).

# Event Label	# Total Hits	# Gen. Roads	# Extracted Roads	Proc. Time
A	928	50	257	2636 ns
B	1008	55	325	3180 ns
C	768	40	111	1468 ns
D	848	45	170	1940 ns
E	608	30	67	1116 ns
F	688	35	90	1300 ns
G	448	20	28	804 ns
H	528	25	41	908 ns

## Data Availability

Not applicable.

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
