# Peer review of "Hough Transform Proposal and Simulations for Particle Track Recognition for LHC Phase-II Upgrade"

_sensors, 2022, doi:10.3390/s22051768_

Round 1

Reviewer 1 Report

Please see enclosed file.

Author Response

Having a hardware tester for the trigger is an excellent idea and should be described in the literature.
However, the English in this paper is poorly written. The authors must improve the punction in the article. For instance, the following sentence in line 27 is very confusing: “In this contest the Trigger [3] circuits are those aimed at selecting the real physical parameters within a huge amount of noisy data.” There should be a comma after content for the sentence to be readable. In addition, the word “contest” is not correct.
That is entirely the wrong word. I believe they mean “After the data are collected”.
These English errors occur throughout the paper and make it very difficult to read. The mistakes are so common that I have difficulty understanding what the authors are saying. Similar English errors must be corrected, so readers do not have to guess what is meant.
Then there are formatting problems. The formulas use subscripts

ANSWER: Thanks for the comments. Now the paper has been completely revised. The main changes have been underlined in yellow and highlighted in blue, and also the punctuation should have been revised in the entire manuscript.
For example, regarding the sentences you mention, these have been changed to: "In this environment, the Trigger circuits select the real physical data within noisy background."
We have also added the figure 4 to better explain the "Forward" and "Backward" processes.

However, the text writes φ0 instead of φ0. It is easy to make the formatting correctly, so the authors should revise the text throughout the paper. The authors should go through the paper and fix similar problems.

ANSWER: yes, right, thanks for the comment. I have updated the formatting style in all the formulas.

The authors should use a spell checker. On line 67, the word Python is misspelled. The full paper should be checked for similar mistakes.

ANSWER: thanks, now it has been corrected

Section 4 (page 155) is what is new in this paper. The paper describes a software implementation of the HT and a VHDL method. There is no comparison of the methods. As a result, it is difficult for the reader to be convinced that the technique is correctly used. There needs to be a comparison.

ANSWERE: Right, now I have reported additional references [18-20] where comparisons between FPGAs and other components have already been studied. The advantage of using FPGAs for low latency applications is emphasized. low latency applications are those needed for high energy physics.

Furthermore, there is a picture of the circuit board. There is no description of the board or its use in practice. There needs to be a discussion on how the board is intended to be used and the experience gained.

ANSWER: yes, the picture of the demonstrator is just used to show that laboratory tests have been done. The reference [16] in particular explains the details of that card, which in facts needs a dedicated paper for a full description. Here the reference to that card is only to show that we are compatible with one of the latest and most powerful cards developed within the big experiments at CERN.

Most of my statements are sketchy. They are because the English of this paper is difficult to understand and needs to be improved for a more comprehensive review significantly.

ANSWER: yes, again, thanks. Now the paper should be more clear and readable.

Reviewer 2 Report

It is not essential but would have appreciated a little more comparison with the current procedures.

Nevertheless, the document is well written and in my opinion is ready for publication.

Author Response

ANSWER: Now the paper has been completely revised. The main changes have been underlined in yellow and highlighted in blue, and also the punctuation should have been revised in the entire manuscript.
We have also added the figure 4 to better explain the "Forward" and "Backward" processes.

Reviewer 3 Report

Dear Authors,

  the reported progress is certainly significant. However, it definitely deserves an improved description. In essence, it lacks clarity in the description of the background knowledge, the implementation, the definition of the metrics for evaluating the results. I will append an edited file with a few remarks. Yellow highlighting means a request for improving the use of the English language or sentences/concepts where clarity has to be significantly improved. Worth investing time to turn it into a good paper.

Author Response

Dear Authors,
  the reported progress is certainly significant. However, it definitely deserves an improved description. In essence, it lacks clarity in the description of the background knowledge, the implementation, the definition of the metrics for evaluating the results. I will append an edited file with a few remarks. Yellow highlighting means a request for improving the use of the English language or sentences/concepts where clarity has to be significantly improved. Worth investing time to turn it into a good paper.

ANSWER: Thanks for the comments. Now the paper has been completely revised. The main changes have been underlined in yellow and highlighted in blue, and also the punctuation should have been revised in the entire manuscript.
We have also added the figure 4 to better explain the "Forward" and "Backward" processes.
References [18-20] are new in this version.

I’m very confused. A histogram for me is a frequency plot but here I do not see anything I can call a histogram. I see the display of what is possibly a set of trajectory in  the HS.
......this shall be the core of the method but I believe it definitely misses clarity

ANSWER: Thanks for the comment. This section now has been fully revised, also section 2.2 that you mention. Not it should be more clear, and what you mention as the core of the algorithms is now described in lines 95-108

Round 2

Reviewer 1 Report

The improvements have made the scientific content acceptable.  Unfortunately, the quality of English is poor.  The English level requires the reader to spend more time than usual to understand what is written.  Here is just one example " Furthermore, in general, it is not possible testing hardware in an emulated HEPE environment before the real system is complete, and that is why some components are validated in advance using simulated demonstrators."  I had to read it several times to discover what it what saying.  The content is valid.  I would recommend the sentence to read:

"Furthermore, as it is impossible to test the hardware before the system is complete, some components need to be validated before assembly."

Such language enables the readers to understand the manuscript much more quickly.

It is my recommendation that authors fully edit the manuscript before publication.

Author Response

Thanks for the comments. The paper has been revised again and the specific comments have been addressed. 

Reviewer 3 Report

Dear Authors,

the paper definitely improved and this can be appreciated. Still some minor text changes, outlined in the attached annotated pdf.

Personally, I would have gone further in the analysis of the results before publication. However, technical details are there and this was the goal of your submission.

best wishes

Author Response

Thanks for the comment. The paper has been again revised according to the overall reviewers' comments.